# Bioactive Compounds of Portuguese Fruits with PDO and PGI

**DOI:** 10.3390/foods12162994

**Published:** 2023-08-08

**Authors:** Diana Farinha, Hélio Faustino, Catarina Nunes, Hélia Sales, Rita Pontes, João Nunes

**Affiliations:** Association BLC3—Technology and Innovation Campus, Centre Bio R&D Unit, Oliveira do Hospital, Rua Nossa Senhora da Conceição nº 2, 3405-155 Coimbra, Portugal; diana.farinha@blc3.pt (D.F.); helio.faustino@blc3.pt (H.F.); helia.sales@blc3.pt (H.S.); rita.pontes@blc3.pt (R.P.); joao.nunes@blc3.pt (J.N.)

**Keywords:** bioactive compounds, high value bioeconomy, PDO, PGI, phenolics compounds, Portuguese fruits

## Abstract

The European Union has established two designations, Protected Designation of Origin (PDO) and Protected Geographical Indication (PGI), to guarantee the authenticity of products with valued specificities associated with the regions where they are produced. The valorization of these products, particularly fruits, not only preserves their natural origins but also promotes the cultivalion of specific regional fruit varieties. This contributes to the preservation of biodiversity and the development of innovative bio-based products in the regions of production. In recent years, substantial efforts have been made to characterize PDO and PGI fruits, particularly in relation to the presence of bioactive compounds with antioxidant and antimicrobial properties. Portugal, with its diverse range of climates and geographical characteristics, is home to numerous fruits with unique flavors, textures, and appearances, many of which are now recognized with PDO or PGI seals. This review compiles data from the production of Portuguese fruits with PDO designations, such as the ‘Maça Bravo de Esmolfe’ (apple) and ‘Pera Rocha do Oeste’ (pear), and PGI designations, such as ‘Citrinos do Algarve’ (citrus)and ‘Cereja da Cova da Beira’ (cherry), and summarizes studies focusing on the bioactive compounds present in these fruits. The presence of bioactive compounds is a significant aspect of nutritious food, associated with health benefits that consumers are increasingly aware of and value.

## 1. Introduction

Since the Paleolithic period, natural food products have been associated with health benefits and the treatment of diseases. With the development of synthetic chemistry, especially since the 19th century, and its huge impact on medicine and food production, the attention from the potential benefits of natural foods was diverted. However, the interaction between animals and fruit plants over millions of years at a global scale has resulted in a large diversity of complex compounds produced by nature, which is hard for man to reproduce in a factory. Therefore, in the last few decades, there has been a renewed interest in foods that goes way beyond satiating hunger, and concerns about the long-term health benefits and potential medicinal and antimicrobial properties are now being considered [1].

Consumed and cherished globally, fruits have been linked to a lower risk of some diseases, lending support to the belief that a high fruit intake might be a preventative measure. This is likely due to the abundant nutrients and energy fruits provide, including vitamins, minerals, fiber, and a myriad of other biologically active compounds. Additionally, fruit crops offer more than just their flesh; the peels, leaves, and barks are also known to harbor medicinal properties. Their collective benefits underscore the importance of fruits in our diets and their potential role in supporting overall health [2].

A diet rich in fruits is recommended to reduce the likelihood of developing diseases [3], and this fact is associated with a large number of bioactive compounds that possess biological functions, such as antioxidant and anti-inflammatory properties, which enhance human health [4].

Bioactive compounds are present not only in the fruit parts used for direct intake, but also in their waste parts; therefore, valuable bioactive compounds are being extracted from the waste of food industries [1]. Over the past few decades, there have been studies and endeavors focused on harnessing and maximizing the potential of beneficial substances found in food waste such as leaves, peels, seeds, and pulp. The aim is to reduce the environmental impact while also extracting valuable compounds that could be commercially viable [5]. The potential of bio-residues from Portugal’s main fruit crops, such as apples, oranges, and pears as sources of functional/bioactive compounds with therapeutic potential, has been recently reviewed [6].

Taking into account the World Health Organization’s recommendations for increased fruit consumption [7], coupled with a growing interest in the valorization of regionally distinct products bearing potential health benefits for the purpose of fostering more sustainable economies, the information pertaining to the bioactive compounds found within these products is becoming progressively more vital to consumers. This trend is evident in the increased frequency with which such information is being displayed on product labels. In light of these considerations, we undertake in this review to collate and analyze studies that focus on the bioactive compounds present in fruits of high economic significance in Portugal.

Fruit growing is an important sector of agriculture in Europe, as it provides millions of tons of fresh and healthy food for consumers, as well as income and employment for farmers and rural communities. According to Eurostat, about 1.3 million hectares of land in the European Union (EU) are covered with fruit trees, accounting for around 1% of the total agricultural area. The main fruit crops grown in Europe are apples, oranges, pears, peaches, and nectarines [8].

In order to guarantee the authenticity of products, the European Union created two designations for the distinction of high-quality products, PDO (Protected Designation of Origin) and PGI (Protected Geographical Indication). PDO refers to a product with a protected designation of origin, meaning it is cultivated or produced in a specific region, and its quality is primarily or exclusively derived from that location. The entire process of production, transformation, and elaboration takes place in the region of origin. On the other hand, PGI refers to a product that is cultivated or produced in a specific region, and its quality is associated with that region. At least one phase of the production, transformation, or elaboration of the product occurs in the area of origin [9].

PDO and PGI labels represent integral aspects of the bioeconomy in relation to Portuguese fruits. These certifications enhance the market value of bio-based products by verifying that their unique characteristics are tied to their geographic origin and increase consumer confidence in their quality and authenticity. This increased profitability bolsters the economic viability of the bioeconomy. Furthermore, the focus of PDO and PGI labels on promoting the cultivation of specific regional fruit varieties contributes to the preservation of biodiversity, a crucial component of a healthy bioeconomy that underpins the resource base for the development of innovative bio-based products.

PDO and PGI designations also play a vital role in sustaining rural economies by supporting local fruit producers [10]. In a bioeconomic context, this can foster rural development and create new bio-based value chains, thereby linking agricultural production with processing industries. Additionally, these labels may drive bio-based innovation by inspiring the development of new, value-added products, such as specialty food items or natural health products, which leverage the unique characteristics of PDO and PGI fruits. The adherence to traditional and sustainable farming practices, often a requirement of these certifications, aligns with the bioeconomy’s emphasis on sustainability and efficient resource use. Lastly, the market recognition and consumer trust fostered by PDO and PGI labels can enhance the visibility of bio-based products and stimulate demand, promoting the growth and success of the bioeconomy.

The purpose of the work was to provide a comprehensive understanding of the importance of bioactive compounds present in Portuguese fruits with POD and PGI designations, further emphasizing the importance of these fruits both for health benefits and economic sustainability. These scientific efforts will undoubtedly provide valuable insights to producers and consumers, enhancing their understanding of the nutritional qualities embedded within these specialty fruits. By highlighting the distinct nutritional advantages that PDO and PGI fruits can offer, producers can enhance consumer awareness and appreciation of the health benefits derived from these specially designated products.

### Fruits Produced in Portugal

Portugal has a favorable climate for many fruits and vegetables development. Over the last few years, with climate alterations, the Portuguese climate has become hotter and has enabled the cultivation of exotic fruits. However, most of the natural products still produced in Portugal are fruits that included in the Mediterranean diet. The FAO (Food and Agriculture Organization of the United Nations), in 2021, reported that the most produced fruits in Portugal were olives (1,375,750 tons), grapes (977,670 tons), apples (368,230 tons), oranges (363,920 tons), pears (225,360 tons), and cherries (23,930 tons) [11].

Portugal is one of the leading countries for citrus and pear production where it ranks in the top 5 European countries with more areas dedicated to citrus fruits and pears [8]. In 2021, Portugal had a total area of 43,802 hectares of land mainly planted with fresh fruit trees, 4353 hectares of land planted with small berry fruits, and 21,681 hectares of land planted with citrus fruit trees [12]. Fruit growing is a significant source of income for many Portuguese farmers, as well as a driver of exports and rural development. However, the sector also faces some challenges, such as labor shortages, climate change, pests and diseases, and market competition. To overcome these difficulties, Portugal needs to invest in innovation, quality, sustainability, and diversification of its fruit products.

Holdings specialized in vineyards are the most common in Portugal, representing around 12% of the total number of Portuguese holdings, while mixed cropping farms account for 10.3%. The holdings specialized in olive production represent 8.5% of the total, a share slightly smaller than those farms dedicated to special fruits and citrus fruits (8.8%). In terms of economic impact, the farms specialized in fruits and citrus fruits production account for 7.7% of the total agricultural holdings.

The cultivation of pears is also a major economic endeavor in Portugal, with an annual production of approximately 190,000 tons. The predominant variety in Portugal is the ‘Rocha’ pear, which is a traditional variety in the country and accounts for 95% of the national pear production. This production is primarily concentrated in the West region of Portugal [13]. In Portugal, there are several fruits with PDO or PGI designation (Table 1), and these fruits are produced from north to south of the country, including the islands of Madeira and the Azores [14].

The percentage of certified fruit production varies greatly depending on the type of fruit (Table 2). For instance, in 2020, Portugal produced a total of 286,075 tons of apples, 17% of which were certified apples (including ‘Maçã Bravo de Esmolfe’ PDO, ‘Maçã da Beira Alta’ PGI, ‘Maçã da Cova da Beira’ PGI, ‘Maçã de Alcobaça’ PGI, ‘Maçã de Portalegre’ PGI, and ‘Maçã Riscadinha de Palmela’ PDO). On the other hand, of the total 9241 tons of cherries produced in the same year, only 3% were certified (including ‘Cereja da Cova da Beira’ PGI, ‘Cereja de São Julião—Portalegre’ PDO, and ‘Ginja de Óbidos e Alcobaça’ PGI). More data about the production of these fruits are presented in Table 2 [15]. Interestingly, while grapes are a major fruit crop in Portugal and contribute significantly to the country’s wine production, the individual grape varieties themselves, such as ‘Touriga Nacional’, ‘Alvarinho’, ‘Baga’, ‘Arinto’, and ‘Encruzado’, do not carry the PDO status. These grape varieties, traditionally grown in specific regions like Douro, Dão, Minho, and Bairrada, are integral to the production of many wines that do carry the PDO status. However, the PDO seal is granted to the wines produced in these regions, not the grape varieties themselves.

## 2. Bioactive Compounds

Bioactive compounds are metabolites used for plants’ defense systems and ecological relationships [1]. Having a direct relationship with the environment, the concentrations of bioactive compounds can vary greatly depending on the location, variety, practices applied to the plants [4], climatic conditions, fruit maturity, and storage [16]. For example, the aromatic compounds of olive oil extracted from olives are highly sensitive to the fruit’s ripening stage, olive tree variety, irrigation conditions, and water stress to which the olive trees are exposed [17].

The most abundant bioactive compounds in fruits and vegetables are carotenoids (such as carotene, lutein, and lycopene) and flavonoids, which include flavonols (myricetin, quercetin, and flavan-3-ols ((−)-epicatechin and (+)-catechin)), flavones (apigenin and luteolin), anthocyanidins (like petunidin, cyanidin, and malvidin), and flavanones (hesperidin and naringenin) [18]. These compounds are known for their potential health benefits and are commonly found in fruits [19].

Bioactive compounds have various functions within the fruits themselves. For example, anthocyanins (polyphenols present in cherry skin) are responsible for the color of the fruits [16], which, in turn, is an attractive characteristic for animals that aid in seed dispersal. However, when extracted and applied in products for human consumption, they are also very beneficial. The health-promoting antioxidants found in food include phenols, tocopherols, phospholipids, amino acids, phytic acid, ascorbic acid, sterols, and carotenoids [20].

Fruits and vegetables are rich in bioactive compounds, but they are not accessible, to obtain them, in many cases, extractions with different solvents are needed. The methodology used in each case depends on the bioactive compounds that one aims to extract from fruits. For the extraction of phenolic compounds, for example, there are several methodologies available, such as conventional solvent extraction, enzyme-assisted extraction, accelerated solvent extraction, microwave-assisted extraction, supercritical water extraction, and supercritical fluid extraction [4].

The bioactive compounds are not only found in fruits and vegetables, but also in the residues/pomace of many food industries. These residues, which, in many cases, are not yet utilized, still contain a significant number of bioactive compounds that can be extracted and used in various industrial applications, thereby reusing and extending the lifespan of these residues, reducing food industry waste [21].

Of all the bioactive compounds, the most investigated are polyphenols or phenolic compounds. These molecules structures consist of at least an aromatic ring directly bonded to one or more hydroxyl groups (–OH) and more than 8000 different structures have been identified. Phenolic compounds span from relatively simple molecules, such as phenolic acids, to more highly complex and intricate structures in polyphenols like condensed tannins. Among the common classification systems for phenolic compounds, a straightforward division usually places them into flavonoids and non-flavonoids categories. Flavonoids, in particular, exhibit a characteristic three-ring structure (C6-C3-C6), comprising two aromatic rings connected by a three-carbon bridge. In nature, they may be associated with sugar molecules, giving rise to glycoside derivatives. Meanwhile, non-flavonoid phenolic compounds comprise an array of structures differing significantly from that of flavonoids [1].

Phenolic compounds are diverse secondary metabolites found in plants. They play a crucial role in determining food characteristics and possess numerous health benefits. These compounds impact color, taste, aroma, and astringency in food [22].

When consumed, they can exhibit biological activities such as anti-inflammatory, antimicrobial, antioxidant, or cytotoxic [1]. Thus, they have an influence on various diseases like vasodilation, reduced vascularization, and improved lipid profiles, which can help regress atherosclerosis. They also lower the risk of thrombosis, myocardial infarction, heart disease, and ischemia. Phenolic compounds exhibit anticancer properties and reduce the risk of diabetes. Overall, they are important phytochemicals with positive effects on both food and human health [22].

### 2.1. Apples

Apples are one of the most popular fruits in the world. More specifically, in Portugal, the traditional apple varieties have gained attention in the last years because of their distinct sensory properties [23]. From the several Portuguese endogenous apple varieties, ‘Bravo de Esmolfe’, ‘Malápio Fino’, ‘Malápio da Serra’, and ‘Pêro Pipo’ stand out [24]. In fact, the demand for the ‘Bravo de Esmolfe’ variety (BE) has been growing [25]. Due to its aroma and flavor properties, this apple was recognized as a ‘Protected Designation of Origin’ product (Diário da República—Despacho 58/94 of 15 February 1994). BE is preferred by consumers for its odor, taste, hardness, and juiciness over other traditional and commercial apple varieties [26].

In general, apples contain 84% water and are rich in minerals such as potassium (K), magnesium (Mg), calcium (Ca), and sodium (Na). They also contain trace amounts of zinc (Zn), manganese (Mn), copper (Cu), iron (Fe), boron (B), fluorine (F), selenium (Se), and molybdenum (Mo). They have a high content of B-complex vitamins, fibers, cellulose, hemicellulose, and lignin [26].

Apples also contain polyphenols, dietary fiber, pectin, triterpenoids, and volatiles [1]. In different apple varieties, the most commonly phenolic compounds are flavan-3-ols and, in smaller quantities, quercetin, (+)-catechin, (−)-epicatechin, procyanidins, phloridzin, and chlorogenic acid [24]. Among Portuguese varieties, ‘Malápio’ Fino is the one that has higher levels of polyphenols such as (+)-catechin, (−)-epicatechin, chlorogenic acid, quercetin-3-glucoside, and procyanidin B1 [24].

In the apple pomace (residue from cider-producing industries), it is possible to find catechins and proanthocyanins, which are the predominant flavanoids found in these residues. In smaller quantities, hydroxycinnamic acid derivatives, dihydrochalcones, and anthocyanins can also be found. The composition of proanthocyanins in apple pomace varies depending on the moisture percentage. In fresh pomace, chlorogenic acid, caffeic acid, (+)-catechin, (−)-epicatechin, rutin, and quercetin glycosides are present in significant amounts. In dry pomace, the most abundant proanthocyanin is phlorizin [1].

#### ‘Maçã Bravo de Esmolfe’PDO (Apple)

The ‘Bravo de Esmolfe’ apple is one of the varieties of the *Malus domestica* Borkh species. It is a medium-sized or small apple, with white flesh that can sometimes have reddish spots, and it has a strong and pleasant smell. This apple variety is well adapted to the mountainous climate, with a late flowering period to survive late frosts. It is produced from Lamego to Fundão regions [27], as shown in Figure 1.

In the ‘Bravo de Esmolfe’ apple, an autochthonous variety from Portugal, fifteen different phenolic compounds were identified, including three epicatechins, two caffeoylquinic acids, two procyanidins, two coumaroylquinic acids, (−)-epicatechin, phloretin-2-*O*-xyloglucoside, and phlorizin [4]. According to Vilas-Boas et al. [28], ‘Bravo de Esmolfe’ is rich in antioxidants and represents one of the most important traditional varieties in Portugal. This study demonstrated that by applying a sustainable extraction technique, it was possible to obtain an antioxidant extract with higher antioxidant activity and polyphenols, including hydroxycinnamic acids (4-caffeoylquinic acid and 5-caffeoylquinic), flavanols (−)-epicatechin, (+)-catechin, and procyanidins), and chalcone (phoretin and phloridzin) [28]. 

Serra et al. [24], tested nine apple varieties, and all of them demonstrated antioxidant and antiproliferative activity. Among them, the ‘Málapio Fino’ variety showed the highest antioxidant activity. ‘Bravo de Esmolfe’ extract has been shown to possess antioxidant and antibacterial activity against both Gram-positive and Gram-negative bacteria [4].

Apples are associated with the reduction in various diseases such as certain cancers (such as prostate, liver, colon, and lung), thrombotic stroke, ischemic heart disease, asthma, and type-2 diabetes [24]. They also contribute to reducing cholesterol levels, human LDL oxidation, and cardiovascular diseases [4].

Among Portuguese varieties, ‘Malápio Fino’ is the one that has higher levels of polyphenols such as (+)-catechin, (−)-epicatechin, chlorogenic acid, quercetin-3-glucoside, and procyanidin B1 [23]. ‘Maçã Bravo de Esmolfe’ (PDO) displays higher values of µmol of Trolox equivalents/100 g in oxygen radical absorbing capacity assay (ORAC) and µmol of caffeic acid equivalents/100 g in the hydroxyl radical adverting capacity assay (HORAC) when compared to more popular varieties such as ‘Golden’, ‘Fuji’, or ‘Gala Galaxy’ [23].

In a comparison of antioxidant capacities with commercially popular apple varieties, ‘Bravo de Esmolfe’ outperforms ‘Golden’, ‘Fuji’, and ‘Gala Galaxy’. With an ORAC value of 1503 µmol of Trolox equivalents/100 g (TE), ‘Bravo de Esmolfe’s’ antioxidant capacity is higher than that of ‘Golden’ (821 TE), ‘Fuji’ (1065 TE), and nearly double that of ‘Gala Galaxy’ (761 TE). Similarly, in the HORAC measurement, ‘Bravo de Esmolfe’ has 796 µmol of caffeic acid equivalents/100 g (CAE), marking more than double that of ‘Golden’ (360 CAE), higher than ‘Fuji’ (551 CAE), and ‘Gala Galaxy’ (446 CAE) (Table 3) [24].

In a dissertation examining ‘Maçã da Beira Alta’ (PGI) (apples), [29] which includes the ‘Golden Delicious’ and ‘Starking’ varieties, as well as the ‘Bravo de Esmolfe’ variety (PDO), it was found that the ‘Bravo de Esmolfe’ variety had significantly higher performance. The pulp ethanolic extract of ‘Bravo de Esmolfe’ contained more than double the content of gallic acid equivalents compared to the ‘Golden Delicious’ and ‘Starking’ varieties, when cultivated in the same region and collected at the same time. This was evident in both total phenolic content and DPPH assays, as indicated in Table 4.

In another dissertation about ‘Maçã Cova da Beira’ (PGI) (apple), using the ‘Bravo Esmolfe’ variety (PDO) [30], a more complete profile of bioactive compounds was described (Table 5). Although the purpose of this study was the evaluation of different extraction methods, ethanolic extract revealed a high value for the ORAC (27.94 μmol CAET/g in the apple) assay.

‘Maçã de Alcobaça’ (PGI) from apple varieties ‘Casa Nova’, ‘Gala’, ‘Granny Smith’, ‘Reinette’, ‘Starking’, ‘Golden’, ‘Fuji’, and ‘Jonagored’ were evaluated by the ABTS•^+^ method. The study reveals a significant antioxidant capacity across all apple varieties, as evaluated by ABTS•^+^ measurements. Interestingly, it was found that the extract of apple peels demonstrated an antioxidant activity approximately five times greater than that of the flesh. The ratio of antioxidant capacity between peel and flesh showed variation among the different varieties. Specifically, the ratio was 3 to 4 times higher in ‘Reinette’, ‘Granny Smith’, and ‘Golden Delicious’, and surged up to 9 times in ‘Starking’. When comparing among varieties, the highest antioxidant activity was detected in the flesh of ‘Reinette’ and the peel of ‘Starking’. Generally, lower antioxidant activity in peel was observed in yellow varieties, such as ‘Golden Delicious’, and green varieties, like ‘Granny Smith’, as opposed to red varieties, ‘bicolor’ and ‘russet’. However, only a comparison is described and no values are provided in the publication [31].

In the apple pomace, it is possible to find catechins and proanthocyanins, which are the predominant flavanoids found in these residues. In smaller quantities, hydroxycinnamic acid derivatives, dihydrochalcones, and anthocyanins can also be found. The composition of proanthocyanins in apple pomace varies depending on the moisture percentage. In fresh pomace, chlorogenic acid, caffeic acid, (+)-catechin, (−)-epicatechin, rutin, and quercetin glycosides are present in significant amounts [1].

### 2.2. Cherry

Cherry, which is a common designation, refers to a group of plant species belonging to the Rosaceae family, specifically the Prunoideae subfamily and *Prunus* genus. These species have their roots in Asia. One prominent species among them is *P. avium* L., commonly known as sweet cherry. Sweet cherries have a global distribution, with a higher concentration in regions characterized by a temperate climate, particularly across Europe [32].

Cherries are rich in natural antioxidant substances called polyphenols, which have been linked to numerous health benefits. These polyphenols found in cherries include various types of flavonoids such as anthocyanins, flavan-3-ols, and flavonols, as well as hydroxycinnamic acids and hydroxybenzoic acids [33].

Anthocyanins are present in cherries like cyanidin-3-glucoside and cyanidin-3-rutinoiside, peonidin-3-rutinoiside, and pelargonidin-3-rutinoside [34]. These compounds slow down cardiovascular diseases and inhibit tumor growth. This fruit, in addition to the mentioned polyphenols, also contains other flavonoids such as quercetin, kaempferol, rutin, (+)-catechin, and (−)-epicatechin, as well as phenolic acids like neochlorogenic acid and chlorogenic acid [16].

Among the hydroxycinnamic acids, the ones that exist in higher concentrations in cherries are caffeoylquinic and coumaroylquinic acids [34], there are also neochlorogenic acid, *p*-coumaroylquinic acid, and chlorogenic acid [35].

In terms of flavanols and flavonols, the most commonly found ones in cherries are (−)-epicatechin and quercetin-3-rutinoside [36]. It is also possible to find rutin [35].

Extracts derived from sweet cherries exhibit remarkable biological potential. This is primarily attributed to their antioxidant properties, which effectively combat free radicals and safeguard cells against oxidative damage. Consequently, these extracts hold promise as potential therapeutic agents for the treatment of inflammatory conditions (such as diabetes, gout, and arthritis), hemolytic anemia, and cancer, as well as neurological and cardiovascular disorders [5,37].

#### ‘Cereja Cova da Beira’ PGI (Cherries)

‘Cereja Cova da Beira’ is a fruit that can come from various cherry tree varieties, such as ‘De Saco’, ‘Napoleão Pé Comprido’, ’Morangão’, ’Espanhola’, and the varieties ‘B. Burlat’, ‘B. Windsor’, and ’Hedelfingen’, which are cultivated in a restricted area known as ‘Cova da Beira’. In this region, which includes the municipalities of Fundão, Belmonte, and Covilhã [38] (Figure 2), the climate is greatly influenced by the proximity to the Gardunha, Estrela, and Malcata mountains, creating a microclimate characterized by cold winters and mild springs. The cultivation of cherries in this specific location results in environmental conditions directly affecting the fruit’s flavor, such as altitude, which is closely related to the cherry trees’ sun exposure.

‘Cereja Cova da Beira’ are characterized by having in their constitution an important number of bioactive compounds, mainly polyphenols such as gallic acid, *p*-coumaric acid, rutin, chlorogenic aid, (−)-epicatechin, cyanidin-3-*O*-glycoside, and quercetin-3-4′-di-*O*-glycoside, the latter being the one that appears in higher concentrations [22]. Vilas-Boas et al. [18], identified that the main phenolic compounds in ‘Cereja Cova da Beira’ included neochlorogenic acid, *p*-coumaric, quercetin-3-rutinoside, and anthocyanins, namely, cyanidin-3-rutinoside [18]. The ‘Cereja Cova da Beira’ demonstrated antioxidant properties and a high total phenolic content. The ORAC value registered at 38.34 mg TE/g, indicating a strong capacity to neutralize free radicals. The ABTS value was determined to be 7.8 mg ascorbic acid equivalent per gram of dry extract (AAE/g), substantiating the cherries’ antioxidative potency. The DPPH value was noted at 6.98 mg TE/g, further confirming the fruit’s antioxidative effectiveness. Moreover, the total phenolic content TPC was measured at 8.75 mg GAE/g, highlighting the high concentration of phenolic compounds and its probable influence on the overall antioxidant potential [19].

Serra et al. [16] confirmed the presence of the aforementioned phenolic compounds and further identified (+)-catechin (flavanols), pelargonidin, and peonidin (anthocyanins). They performed the extraction of phenolic compounds using a hydro-methanolic solution and conducted several tests to assess the biological activities of the extract in nine varieties of cherries. The study confirmed that the extract obtained from cherries from the Cova da Beira region, specifically the ‘Morangão’ and ‘Saco’ varieties, displayed antioxidant activity. Among the tested varieties, the ‘Saco’ variety exhibited the highest antioxidant capacity (Table 6). Moreover, the study also revealed the extract’s antiproliferative activity against HT29 (human colon cancer cells) and MKN45 (human gastric cancer cells) [16].

The PGI, ‘Ginja de Óbidos e Alcobaça’, refers to the sour cherries grown in a traditional area of Portugal. This region is located between the Candeeiros and Montejunto mountains, extending all the way to the Atlantic Ocean. The geographical location of this region creates a unique microclimate. When coupled with the area’s fertile soils, this microclimate enables the growth of sour cherries that have distinct chemical and sensory (organoleptic) characteristics when compared to sour cherries grown elsewhere. An interesting morphological feature of the sour cherries grown in this region is ‘Folha no Pé’. This refers to a small growth, about 1 cm long, where the stems of the flowers and fruits are attached. On this small growth, two to four small leaves appear. This distinctive feature contributes to the uniqueness of the ‘Ginja de Óbidos e Alcobaça’. Due to their unique characteristics and the geographical factors contributing to their growth, these sour cherries are used to produce highly appreciated products. The most famous among these are the liqueurs and sweets, which are renowned for their exceptional taste and quality [39].

The ‘Folha no Pé’ and ‘Galega’ sour cherries varieties both exhibit antioxidant capacities and phenolic compound concentrations. The antioxidant capacity of the ‘Folha no Pé’ cherries was measured at 189.2 mg of ascorbic acid equivalent per 100 g of fresh weight, while the ‘Galega’ cherries exhibited a slightly lower value of 147.36 mg of ascorbic acid equivalent per 100 g. These results suggest a substantial potential for free radical scavenging in both varieties. Further, the TPC was also measured, with the ‘Folha no Pé’ cherries registering at 823.3 mg GAE/100 g and the ‘Galega’ cherries at 598.0 mg GAE/100 g PF. This underscores the high concentration of phenolic compounds in these cherries, which likely contributes significantly to their overall antioxidant potential [39].

Most of the production of ‘Ginja de Óbidos e Alcobaça’ goes to the production of Sour Cherry Liquor, leaving a pomace (mainly constituted by the skin and kernel). High-performance liquid chromatography (HPLC) facilitated the identification and quantification of phenolic compounds including cyanidin-3-*O*-glucoside, (+)-catechin, (−)-epicatechin, and several phenolic acids. These compounds have the potential for integration into nutraceutical formulations or for utilization within the food or cosmetic sectors, thereby expanding their potential applications and value [40].

### 2.3. Olives

Table olives are important components of the Mediterranean diet and are largely consumed in the world. In Portugal, concerning table olives and olive oils, eight products are already registered as PDO, one of them being ‘Azeitonas de Conserva de Elvas e Campo Maior’ (canned olives) [41].

Olives have a significant number of phenolic compounds, with different degrees of complexity. Among the simple compounds, we can find benzoic or cinnamic acids, glycosylated or free flavonoids (primarily based on luteolin, quercetin, or apigenin), and phenolic alcohols such as tyrosol and hydroxytyrosol. But more complex phenolic compounds, including verbascoside, which is a combination of caffeic acid, rutinose, and hydroxytyrosol, can also be found. In addition to these compounds, olives also contain phenolic compounds exclusive to olives, namely, oleuropein and demethyloleuropein, which have a mixed biosynthetic origin from shikimate and mevalonate [42].

Pereira et al. [40] performed an analysis of phenolic compounds that was performed by reversed-phase HPLC/DAD, and seven compounds were identified and quantified: hydroxytyrosol, tyrosol, 5-*O*-caffeoilquinic acid, luteolin 7-*O*-glucoside, rutin, and luteolin [41]. The natural antioxidants polyphenols are one of the main olive secondary metabolites, representing about 2–2.5% of the pulp [40].

Olives are another example of how pomace from olive oil production can be used for extracting bioactive compounds. Olive pomace is rich in dietary fiber, unsaturated fatty acids, minerals, and phenolic compounds. In a liquid fraction of olive pomace, it was possible to extract hydroxytyrosol (which helps prevent various diseases such as cancers and digestive disorders), phenolic compounds (such as caffeic and *p*-coumaric acids, and luteolin), and tyrosol. This liquid fraction of olive pomace has demonstrated antioxidant and antimicrobial activity [21].

#### 2.3.1. ‘Azeitona Galega da Beira Baixa’ PGI (Olive)

‘Azeitona Galega da Beira Baixa’ is the processed fruit of the *Olea europaea* L. olive tree species and can be transformed into canned or table olives. These olives have an average weight of 2.5 g and can have a black or sepia-brown color, with a slight wine-like aroma. As with all PGI products, the local region imparts unique characteristics to the product. In the case of ‘Azeitona Galega da Beira Baixa’, the climate and soil have the greatest influence. This type of olive is produced in various municipalities of Beira Baixa, such as Covilhã, Belmonte, Fundão, Penamacor, Idanha-a-Nova, Castelo Branco, Vila Velha de Ródão, Proença-a-Nova, Oleiros, Sertã, Vila de Rei, and Mação [43] (Figure 3).

#### 2.3.2. ‘Azeitona de Conserva de Elvas e Campo Maior’ PDO (Canned Olives)

The ‘Azeitona de Conserva de Elvas e Campo Maior’ (canned olives) are fruits from the olive tree (*Olea europaea*), and they can be of four different varieties: ‘Azeiteira’, ‘Carrasquenha’, ‘Redondil’, and ‘Conserva de Elvas’. These varieties produce little oil, but the olive trees yield a high number of olives per year. Many of the characteristics associated with this fruit come from the specific location where they are produced, such as the mild local temperature (22 °C), the rainfall associated with Elvas, and the deep and fertile soils of the region. The cultivation of this canned fruit is exclusive to the municipalities of Elvas and Campo Maior [44] (Figure 4).

There have been limited studies on the compounds and biological activities of olives due to a significant portion of this raw material being used for olive oil production.

Vinhas et al. [45] identified by HPLC the phenolic compounds of olives collected in the North and Center of Portugal. These olives contained flavonoid and non-flavonoid phenolic compounds. Among the non-flavonoid phenolic compounds, the olives possess hydroxytyrosol, oleuropein, 5-caffeoylquinic acid, and verbascoside. In terms of flavonoidic compounds, they contain cy-3-glucoside, cy-3-rutinoside, lut 7-gluc, rutin, api 7-gluc, quer 3-rham, and luteolin [45].

J. Pereira et al. [41] analyzed the composition of phenolic compounds, antioxidant potential, and antimicrobial activity in various types of Portuguese table olives that have a Protected Designation of Origin (PDO). The olives studied included natural black ‘Galega’ olives (G-NBO), black ripe ‘Negrinha de Freixo’ olives (NF-BRO), ‘Azeitona de Conserva Negrinha de Freixo’ olives (NF-PDO), and ‘Azeitona de Conserva de Elvas e Campo Maior’ olives (CE-DO) [41]. In this study, DPPH scavenging was calculated through the discoloration of a DPPH solution at a concentration of 6 × 10^−5^ mol/L. The EC50 (half-maximal effective concentration) of the various concentrations of table olive extracts was calculated. The EC50 values obtained for the different olives were G-NBO, 0.47 mg/mL; NF-BRO 0.94 mg/mL; NF-PDO 0.64 mg/mL; CE-DO 0.60 mg/mL, with all the samples exhibiting a considerable capacity at low concentrations, with the PDO NF-BRO variety standing out. In the same study, the antimicrobial activity of different table olives from Portugal was evaluated against several microorganisms. The response for each microorganism tested was different, with olives produced according to the traditional process (NF-PDO and CE-DO) showing the best results. CE-DO extracts inhibited all the tested microorganisms, with the exception of *B*. *subtilis* (Gram-positive). Olive extracts from NF-PDO exhibited considerable growth inhibitions for *B. cereus*, *B. subtilis*, *P. aeruginosa*, *E. coli*, *K. pneumoniae*, and *C. neoformans*. However, the same olive variety (‘Negrinha de Freixo’) submitted to other treatments (Californian-style) only inhibited *B. cereus*, *B. subtilis*, and *E. coli* growth, and with a lower capacity. *B. cereus* and *E. coli* were the most sensitive microorganisms, being inhibited by all the extracts tested. On the other hand, *C. albicans* was the most resistant, only being susceptible to the CE-DO sample.

A recent work by A. Barros [46] evaluated the phytochemical composition and antioxidant capacity of olive seed extracts from different varieties (‘Cobrançosa’, ‘Galega vulgar’, and ’Picual’) from a certified olive grove located in Elvas [46]. As a residue from olive oil, olive seed extracts may provide a sustainable source of bioactive compounds for medical and industrial applications. In this study, the methanolic extract of seeds from ‘Galega vulgar’ outperform ‘Picual’ and ‘Cobrançosa’ in TPC, ABTS, and DPPH assays (Table 7).

Specifically, when focusing on the ‘Galega vulgar’ olives from Beira Baixa, Elvas, and Campo Maior, the available information is even more scarce. However, there are numerous articles on the olive oil extracted from these varieties, and if the oil contains bioactive compounds, it is highly likely that they originate from the olives themselves.

Thus, Peres et al. [47] analyzed the phenolic compounds of extra virgin olive oil from ‘Galega vulgar’ olive trees from Beira Baixa and found different tocopherols, hydroxytyrosol, vanillic acid, vanillin, *p*-coumaric acid, luteolin, and apigenin. The concentration of these compounds in the oil varied when analyzing oils from different olive groves and during different months of olive harvesting (October and November) [47].

More recently, Martins et al. [48] characterized the phenolic compounds present in extra virgin olive oil from various varieties in the Alentejo region. The study identified a total of 107 compounds, primarily belonging to the chemical classes of esters, alcohols, aldehydes, acids, ketones, linear and branched alkyl sulfur compounds, and terpenoids. The ‘Galega vulgar’ variety is known for its significantly low levels of acidic compounds, as well as compounds like aldehydes, terpenes, ketones, and sterols groups.

### 2.4. Citrus

Citrus fruits are origin in the family of *Rutaceae* and believed that the first time they were cultivated was 4000 years ago on the Asian continent [49]. Citrus including orange, mandarin, lemon, lime, and grapefruit are widely cultivated fruits worldwide.

Orange is certainly one of the most consumed fruits in the world within the citrus family. In 2019, approximately 79 million tons of *C. sinensis* (common orange) were produced worldwide. Oranges are consumed in many different forms, such as fresh fruit, juice, preserves, pulp concentrates, and processed foods, among others, which contributes to their popularity [20].

These fruits have gained significant commercial importance due to their abundance of phytochemicals and bioactive compounds [5]. The citrus family has been gaining an increasing number of consumers in recent years, not only due to their attractive appearance, pleasant fragrance, and diverse range of flavors from sweet to sour, but also because they are increasingly associated with health benefits for those who consume them [20].

Citrus fruits are rich in vitamin C, folic acid, potassium, pectin, and other beneficial compounds such as flavonoids and phenolic acids [49]. These compounds have the potential to promote good health. Research suggests that selective consumption of citrus fruits, which contain flavonoids, can help prevent certain types of cancer. Flavonoids, specifically hesperidin and diosmin found in citrus fruits, have demonstrated various therapeutic properties, including antihypertensive, diuretic, analgesic, and hypolipidemic activities. Additionally, citrus fruits contain significant amounts of hydroxycinnamic acids (HCA) such as ferulic, *p*-coumaric, sinapic, and caffeic acid [50].

Approximately 50% of orange fruits are used for juice processing. This generates significant amounts of by-products, including peels, seeds, and pulps, which contain numerous bioactive compounds. Citrus molasses, terpene oils, citric acid, pectins, and essential oil are the main products recovered from orange processing waste [51].

Orange peels generally have 50–100% higher levels of phenolic compounds compared to the pulp or juice [20]. The orange peels are rich in phenols, particularly hydroxycinnamic acids, flavanone (naringin and hesperidin), and polymethoxyflavones [20,49]. These compounds exhibit antioxidant, anti-inflammatory, anti-obesity, anti-aging, anti-diabetic, anti-tumor, anticancer, and even antiviral activities [20]. The essential oil extracted from orange peels contains a mixture of volatile compounds such as terpenes and oxygenated derivatives, including aldehydes, alcohols, and esters. All of these compounds can be extracted and repurposed for inclusion in new products [51].

#### ‘Citrinos do Algarve’ PGI (Citrus)

The ‘Citrinos do Algarve PGI’ fruits are sweet citrus fruits with a thin skin and a vibrant, intense color. The citrus region and associated climate allow for two peak production periods each year, in December/January and June/July. These fruits are cultivated in the southern region of Portugal in municipalities such as Albufeira, Castro Marim, Farim, Faro, Lagoa, Lagos, Monchique, Olhão, Portimão, S. Brás de Alportel, Silves, Vila Real de Santo António, Loulé (excluding the parish of Ameixial), and Tavira (excluding the parish of Cachopo) [52] (Figure 5).

‘Citrinos do Algarve’ with PGI designation is composed of all fruits derived from the following species: *C. sinensis* (oranges), *C. reticulata*, *C. unshiu*, *C. deliciosa*, *C. mobilis* (small citruses), *C. paradisi* (grapefruits), *C. limon* (lemons), *C. limetta*, and *C. limettioides* (limes) [52]. There are no studies where ‘Citrinos do Algarve’ are the subject of study, but there are several works on the bioactive compounds present in different species of citrus.

*C. sinensis*, or oranges, are rich in phenolic compounds such as benzoic acids (like gallic acid, protocatechuic acid, chlorogenic acid, *p*-hydroxybenzoic acid, vanillic acid, and syringic acid) and cinnamic acid (like caffeic acid, *p*-coumaric acid, ferulic acid, sinapinic acid, and cinnamic acid), flavanone glycoside (such as hesperidin, naringin, rutin, and phloridzin), flavonol (myricetin, quercetin), flavanone (hesperetin, naringenin, and kaempferol), and flavone non-glycoside such asluteolin [53].

Lemon (*C. limon* Burm.) is the third most cultivated *Citrus* species in the world and is widely used as a fresh fruit in beverages, as a culinary ingredient, and as a food preservative. Lemon is extensively utilized in the food industry, where it is processed and transformed into jam, drinks, ice cream, and desserts, and it is also used for extracting essential oils [54]. Inside phenolic acids, the most important bioactive compounds present in *C. limon* fruit and juice are flavonoids, such as eriodictyol, hesperidin, hesperetin, naringin (flavanones), apigenin, diosmin (flavones), quercetin (flavonol), and their derivatives. In addition to these, other flavonoids like limocitrin and spinacetin (flavonols) and orientin and vitexin (flavones) are also found. Some flavonoids, such as neohesperidin, naringin, and hesperidin, are specific compounds of lemons. Compared to other *Citrus* species, *C. limon* has the highest content of eriocitrin. In addition to flavonoids, *C. limon* also contains phenolic acids such as dihydroferulic acid, *p*-hydroxybenzoic acid, 3-(2-hydroxy-4-methoxyphenyl) propanoic acid, and synapic acid [55].

In a comparison of antioxidant capacities of orange juice from different varieties in the Algarve region through tests such as TEAC (Trolox equivalent antioxidant capacity) and ORAC (Oxygen Radical Absorbance Capacity), the ‘Valencia Late’ variety from Faro shows the highest antioxidant power in this technique, with a value of 2.00 mol of Trolox equivalents/L. In terms of ORAC, the ‘Encore’ from Faro, followed closely by the ‘Navelate’ from Silves, with values of 3.45 and 3.40 mol of Trolox equivalent/L (TE/L), respectively, are the two varieties with the highest antioxidant power [56] (Table 8).

### 2.5. Pears

Pears are the fruits of trees belonging to the species *Pyrus Communis* L. They are the second most produced and consumed stone fruit in the world (FAO, Roma, Italy) [11]. This fruit can be consumed fresh, processed into syrups, preserves, and purees for use in nectars and yogurts, and, more recently, dehydrated [57].

During growth and development, fruits accumulate organic acids, simple sugars, starch, dietary fiber, antioxidants, minerals, vitamins, and other nutritional compounds, which define the composition and nutritional quality of the fruit [58].

As previously mentioned, the quality and quantity of bioactive compounds present in pears also vary depending on the variety. In pears, sugars, organic and fatty acids, amino acids, phenolics, vitamins, volatiles, and minerals can undergo changes, thereby impacting the aroma and taste of pears [13].

Pears have various compounds that confer them numerous biological activities. Phenolic compounds like chlorogenic acid contribute to reducing inflammation and increasing vasodilation and cognitive function. Arbutin enhances skin whitening, caffeic acid promotes neuroprotection, minerals and vitamins aid in ionic homeostasis, and flavonoids reduce reactive oxygen species (ROS), alleviate allergies, and decrease inflammation. Triterpenoids help combat obesity, diabetes, and hepatic steatosis. Malaxinic acid improves blood circulation, and the fiber and sorbitol present in pears help combat obesity and have a prebiotic action [59].

#### ‘Pera Rocha do Oeste’ PDO (Pear)

The ‘Pera Rocha do Oeste’ PDO (pear)is a fruit from the Rosaceae family, subfamily Pomoids. It is a soft fruit with white and granular flesh, sweet, non-acidic, very juicy, and with a slight aroma. It is cultivated using traditional methods in regions with suitable climates (requires a cold climate). The typical characteristic of these pears is the ‘carepa’ (splatter pattern on the skin), which is concentrated at the base and apex of the pear. The ‘Rocha’ pear variety is cultivated from Pombal to Sintra [60], as can be seen in Figure 6. The municipalities of Bombarral, Cadaval, Caldas da Rainha, and Lourinhã are the main areas known for their extensive pear orchards, covering approximately 70% of the total cultivated area and production [61].

‘Pera Rocha do Oeste’ PDO is characterized by its firmness, acidity, soluble solids content, color, high digestibility, nutritional value, and abundant content of antioxidant phenolic compounds [62]. The pear’s skins, seeds, and flesh are composed primarily of sugars, fibers, pectins, other insoluble carbohydrates, minerals, organic acids, and phenolic compounds [62]. In 2019, 60% of the Portuguese production of ‘Rocha’ pear was exported [63].

The composition of total phenolic compounds in this variety is consistent with the value for other types of pears, with values ranging between 39.3 and 93.1 and averaging 66.2 mg GAE/100 g. Additionally, the vitamin C content was found to vary between 2.1 and 10.4 mg ascorbic acid/100 mL. This range is characteristic of this specific pear variety, which is known for its relatively low vitamin C content. The study includes different ripening stages (ripening index between 33 and 140) and refrigerating conditions; however, no influence of those parameters was observed in total phenolics or vitamin C content [61].

In a study carried out by Salta et al. [13], the phenolic composition and antioxidant potential of the Portuguese pear variety ‘Rocha’ were analyzed and compared with the profiles of commercially prevalent pear varieties, namely, ‘Comice’, ‘Abate’, ‘General Leclerc’, and ‘Passe Crassane’. ‘Rocha’ pear exhibited a superior concentration of total phenolics, with the predominant constituents being identified as chlorogenic, syringic, ferulic, and coumaric acids, as well as arbutin and (−)-epicatechin (Table 9). Furthermore, ‘Rocha’ pear outperformed the other varieties in antioxidant assays, specifically in the DPPH radical scavenging activity (Table 9, lowest IC50) and ferric-reducing power tests [13,62].

Santos et al. [64] study fresh and dried ‘Rocha’ pears from five distinct geographical locations: Torres Vedras, Azueira, Bombarral, Cadaval, and Caria. The first four locations fall within the region of Protected Designation of Origin (Região do Oeste), while the final location, Caria, is situated in a separate region known as Beira Baixa. This study reveals that there are no significant differences between the properties of the pears from the different locations. It was also concluded that the concentration of phenolic compounds decreased after the pears were dried, particularly in the peels (Table 10). Since antioxidant activity can be related to phenolic compounds, this parameter also changed after drying, with dried pears showing a decrease in antioxidant activity (Table 10).

### 2.6. Almond

Almonds began to be cultivated in the fertile crescent region of the middle east 5000 years ago. Almonds are taxonomically close to peaches. Almonds contain nutrients such as proteins, lipids, carbohydrates, and sugars (with sucrose representing 90% of the sugars found), minerals (potassium, calcium, magnesium, phosphorus, and manganese), tocopherols, and riboflavin (vitamin B2) [65].

The almond kernels contain various phenolic compounds, with the highest concentrations being (+)-catechin (2.67–117.59 mg/kg), chlorogenic acid (0.47–21.03 mg/kg), and naringenin (0.05–14.49 mg/kg). Other phenolic compounds present in smaller amounts include hydroxybenzoic acid, gallic acid, rutin, apigenin, and kaempferol. In almonds, 70–100% of the phenolic compounds are present in the fruit’s skin, where compounds such as simple phenolics (hydroxybenzoic and hydroxycinnamic acids, and stilbenes), flavonoids, tannins, and proanthocyanidins can be found. The most abundant compound in the skin is chlorogenic acid [65].

The by-products of almond processing, like shell, skin, and blanch water, contain, as previously mentioned, numerous phenolic compounds that can be recovered and possess biological activities [66]. An et al. [67], found that hydroethanolic almond hull extracts rich in polyphenols have antioxidant properties. These extracts were shown to inhibit the toxicity caused by oxidative stress in a Caco-2 cancer cell line by scavenging reactive oxygen species (ROS) and regulating the redox status of the cells.

#### ‘Amêndoa Douro’ PDO (Almond)

‘Âmendoa Douro’ are the fruits of various varieties of the species *P. amygdalus* L. Almonds are dried fruits with a uniform color and refined properties thanks to the climate and soil of the region where they grow. From the green fruit to the moment of consumption, almonds go through several processes such as crushing, separation of the shell and kernel, calibration, blanching, and roasting. The final product is clean almonds with characteristic color, odor, and flavor. The production area of ‘Amêndoa Douro’ extends from the municipalities of Alfândega da Fé to Figueira Castelo Rodrigo [68] (Figure 7).

‘Amêndoa Douro’ is an almond PDO variety grown in the Douro region of Portugal, known for its rich soils and favorable climate, which contribute to the distinct characteristics of the almonds. The literature studies about ‘Amêndoa Douro’ are very limited. In a study comparing ‘Amêndoa Douro’ and commercial non-PDO varieties, the tocopherol and tocotrienol analysis, reveals that the mean values of all vitamers, except α-tocotrienol, did not show significant differences among the assayed varieties. Interestingly, α-tocotrienol content was found to be greater in PDO varieties, suggesting that specific genetic characteristics might influence the amounts of each tocopherol homolog. The study also found that α-tocopherol was the major compound followed by γ-tocopherol, while δ-tocopherol was the minor vitamer in all varieties. Despite these findings, the results did not reveal potential to discriminate between PDO and commercial varieties based on their tocopherol and tocotrienol content [69].

### 2.7. Annona Genus

The genus *Annona* comprises approximately 100 species that can be found in America, Asia, Africa, and Southern Europe. The fruit is sweet and rich in carbohydrates, thiamine, riboflavin, niacin, iron, calcium, and phosphorus. It contains various bioactive compounds such as alkaloids, flavonoids, glycosides, saponins, carbohydrates, proteins, phenolic compounds, and amino acids [70].

Due to the presence of phenolic compounds in *Annona* fruits, extracts from this fruit exhibit antioxidant activity (in the fruit), antimicrobial activity (in essential oils from fruits), as well as antitumor and antiparasitic properties (in seeds) [70].

Barreca et al. [71], tested three extracts (methanol, ethanol, and dimethylformamide) from the pulp of *A. cherimolia* Mill. Among the three, the ethanol extract showed the highest content of phenolic compounds, with 6.83 µM (GA/1 µL of extract) of total phenols and 5.72 µM (EC/1 µL of extract) of total flavonoids. Regarding proanthocyanins, the extract with the highest quantity was the dimethylformamide extract, which presented 4.31 µM (EC/1 µL of extract). This latter extract also showed the highest inhibition of radical scavenging activity using the DPPH method, with 70% inhibition, while the ethanol and methanol extracts only exhibited 50% inhibition. The dimethylformamide extract also demonstrated the highest radical scavenging activity using the ABTS method, with a value of 35% inhibition.

Shehata et al. [72], tested extracts from two different species of *Annona* and found that in both species, the seeds had higher antioxidant power compared to the peels and pulp. They further tested these extracts on cancer cells and observed that the *Annona* extracts induced apoptosis (cell death) in various cancer cell lines, including Caco2 (colorectal), PC3 (prostate), HepG-2 (liver), and MCF-7 (breast) cells. These findings suggest that the polyphenols present in the extracts may possess anticancer properties and could potentially be beneficial in the treatment or prevention of certain types of cancer.

#### ‘Anona da Madeira’ PDO (Cherimoya)

The PDO ‘Anona da Madeira’ is a fruit belonging to the species *A. cherimolia* Mill. It is a delicate exotic fruit with two varieties: ‘Anona Lisa’ and ‘Anona de Escamas’. ‘Anona Lisa’ is named as such because its skin is smooth and U-shaped. It has a dense, firm, and sweet pulp with few seeds. On the other hand, ‘Anona de Escamas’ has scales on its skin and a thick peel. Its pulp is juicy and white, with many seeds. This variety is highly appreciated. PDO ‘Anona da Madeira’ is cultivated on the island of Madeira [73] (Figure 8).

The ‘Anona da Madeira’ is the second most exported fruit from the Autonomous Region of Madeira. This fruithas Protected Designation of Origin status since 2000, making it the first fruit at the regional level to receive this degree of international protection. It is a highly appreciated fruit with distinct organoleptic and nutritional characteristics, but it is still very under-studied regarding its nutritional value and bioactive compounds. A brief study on the antioxidant activity, phenolic compounds, and total flavonoids of four varieties of *A. cherimola* Mill. from Madeira (Funchal, Madeira, Mateus II e Perry Vidal) is present. The results of the DPPH assay indicate that the ‘Annona da Madeira’ variety presented the lowest EC50 values for all parts of the fruit, thus demonstrating the highest antioxidant activity. The content of phenolic compounds was quite similar among all varieties, although there were considerable differences at the level of the different parts of the fruit. The peel presented the highest content of phenolic compounds, which varied between 17.0 and 19.6 mg of GAE 100 g of sample for the ‘Perry Vidal’ and ‘Mateus II’ varieties, respectively [74].

### 2.8. Peach

The peach fruit can be divided into three distinct parts. The first part is the juicy and yellow pulp, also known as the mesocarp, which makes up approximately 75.2% of the fruit’s total weight. The pulp offers a delightful combination of flavors, with a balance of acidity and sweetness that can vary significantly. The second part of the peach is the peel, or exocarp, which accounts for around 22.5% of the fruit. Lastly, the peach has a stone, which is the endocarp. This stone holds the seed, enclosed within a hard shell called the seed shell or kernel shell. The seed itself represents approximately 5 to 12.5% of the fruit’s weight, depending on the specific peach variety [75].

Peaches are known for their nutritional value and therapeutic properties [76]. In peaches, it is possible to find different flavonoids such as procyanidin, (+)-catechin, quercetin, rutin, quercetin, and chlorogenic acid [77], and carotenoids (β-carotene, lutein, and zeaxanthin) [78].

In the *P. persica* species, phenols and flavonoids can be found, with the highest concentration in the pulp, while carotenoids are more concentrated in the seeds. In terms of antioxidant activity, seed extracts demonstrate a higher value of antioxidant activity, measured by the DPPH method (85.3 µL/mL) [76].

The peach industry utilizes different processes depending on the desired end product. The most common products include peaches preserved in syrup, either in cans or glass containers, as well as concentrated peach purée. The peach purée is a versatile ingredient used in various food formulations, such as baby food, juices, jams, fruit pulp, and yogurt [79]. The by-products generated during peach processing, such as pomace, seeds, and seed shells, are highly valuable and renewable biomasses. These by-products contain various beneficial components, including lipids, proteins, phenolic compounds, fibers, and important bioactive phytochemicals such as neochlorogenic acid, gallic acid, caffeic acid, and procyanidin. These compounds are associated with several health benefits, including anti-inflammatory, anti-obesity, and anti-cerebral ischemia properties [75].

#### ‘Pêssego da Cova da Beira’ PGI (Peach)

The ‘Pêssego da Cova da Beira’ PGI, derived from various varieties of *P. persica* Sieb. and Zucc. species, is characterized by its succulent yellow flesh and rich flavor. The fruit’s production, driven by regional peach tree varieties ‘Dixired’, ‘Red Top’, ‘J. H. Hale’, ‘Merril Franciscan’, ‘Black’, Rubidoux’, ‘Carnival’, and ‘Halloween’, is facilitated by well-drained granitic mid-hill soils and a unique climate influenced by the location between the Gardunha, Estrela, and Malcata mountains. This environment provides ample cold, a mild spring, and protection against Atlantic winds, leading to a peach distinctly different from those in other regions. Each variety of peach produced in this region has unique color and pigmentation, dictated by a sugar content always exceeding 7% [80] (Figure 9).

In a dissertation about ‘Pêssego da Cova da Beira’ as health promotors using six peach varieties (‘Summer Rich’, ‘Fidelia’, ‘Royal Glory’, ‘Royal Magister’, ‘Royal Lu’, and ‘Sweet Dreams’) from the Fundão region (Portugal), DPPH assays showed that extracts exhibited antioxidant activity in a concentration-dependent manner. The most active was ‘Royal Lu’ with an IC50 = 62.0 µg/mL, followed by ‘Sweet Dreams’ (IC50 = 65.1 µg/mL) and ‘Fidelia’ (IC50 = 79.0 µg/mL). All extracts were more active than the positive control, ascorbic acid (IC50 = 18.9 µg/mL). ‘Royal Lu’s’ activity was approximately three times superior to that of the ascorbic acid, and the least active variety, ‘Summer Rich’ (IC50 = 146.8), was seven times higher than the positive control. The antioxidant activity of the peach extracts is mainly due to their phenolic contents, emphasizing phenolic acids and considerable amounts of cyanidin-3-*O*-glucoside, both with high hydrogen donating capacity due to their structures rich in hydroxyl groups [81].

## 3. Summary and Future Perspectives

The study of phenolic compounds present in each fruit, or even in the residues resulting from their industrial use, is very important, not only because they are bioactive compounds with biological capabilities, but also because when extracted, they can still be reused, thus extending the shelf life of the fruits or their residues. The extracted bioactive compounds are a valuable resource for the discovery of new molecules that can be used in the food, pharmaceutical, or cosmetic industries.

There are few studies on the bioactive compounds present in Portugal’s endemic fruits. This knowledge is extremely important, as it will not only enhance the value of the fruits but also help develop the regions where the fruits are produced.

This article aimed to gather information about the PDO or PGI fruits of Portugal. However, no information was found regarding the ‘Ananás dos Açores/São Miguel’ (pineapple) and the ‘Meloa de Santa Maria—Açores (melon). It is important for future research to determine the compounds present in these fruits. Moreover, expanding the existing information on the other fruits is crucial, as in many cases, the available information is scarce.

## Figures and Tables

**Figure 1 foods-12-02994-f001:**
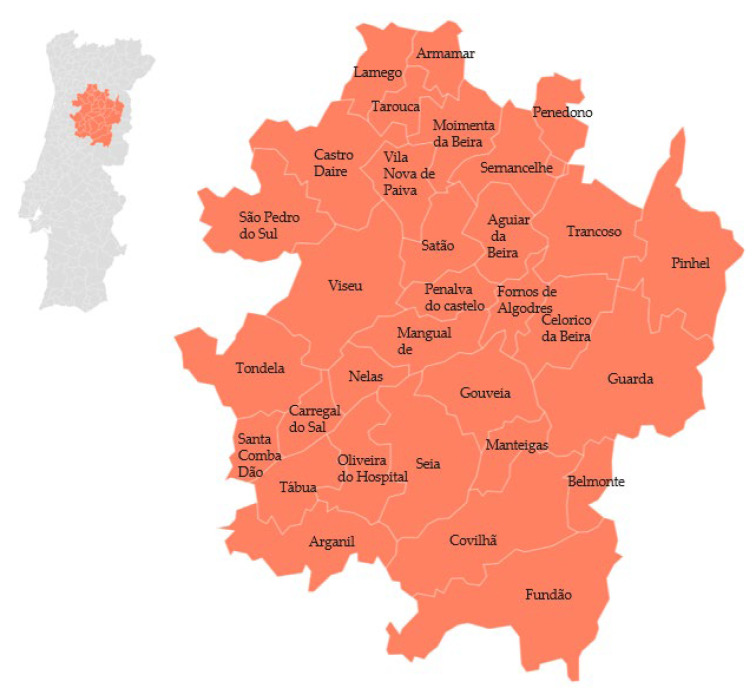
Geographical distribution of ‘Maçã Bravo de Esmolfe’ PDO (apple). Adapted from [27].

**Figure 2 foods-12-02994-f002:**
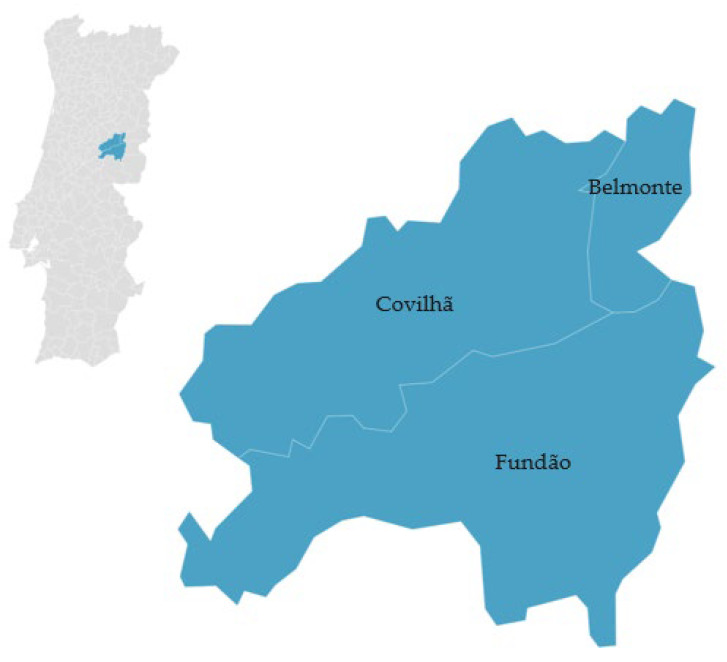
Geographical distribution of ‘Cereja Cova da Beira’ PGI (cherries). Adapted from [38].

**Figure 3 foods-12-02994-f003:**
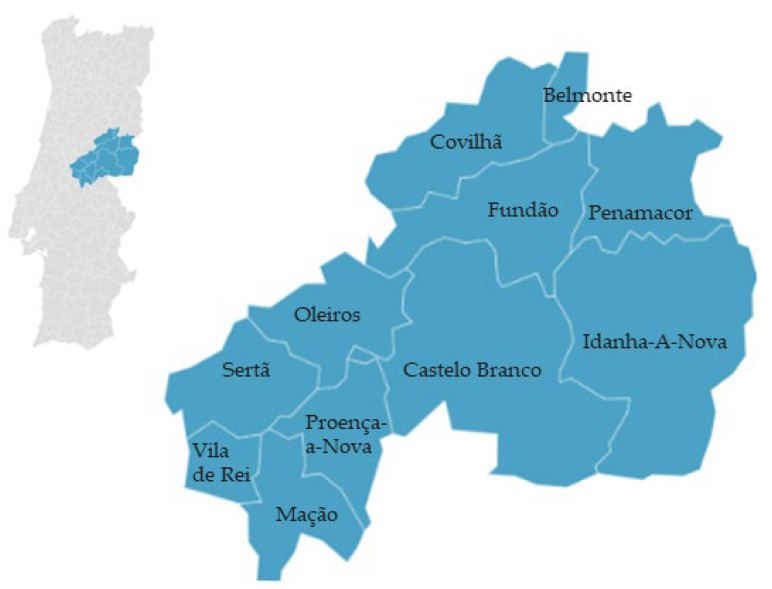
Geographical distribution of ‘Azeitona Galega da Beira Baixa’ PGI (olives). Adapted from [44].

**Figure 4 foods-12-02994-f004:**
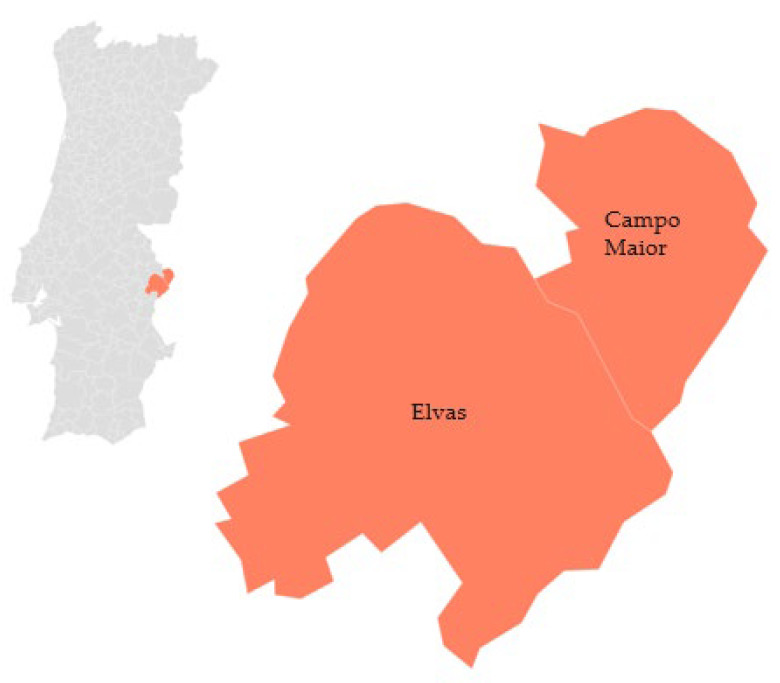
Geographical distribution of ‘Azeitona de Conserva de Elvas eCampo Maior’ PDO (canned olives). Adapted from [45].

**Figure 5 foods-12-02994-f005:**
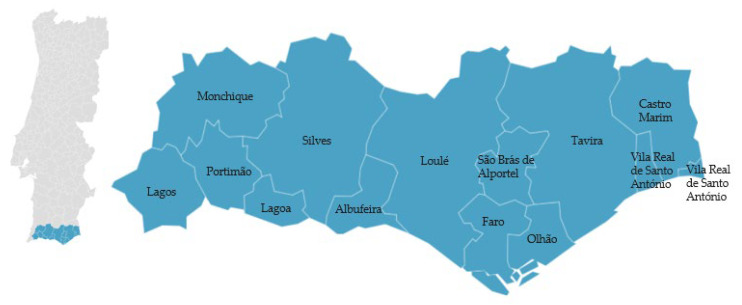
Geographical distribution of ‘Citrinos do Algarve’ (citrus fruits). Adapted from [52].

**Figure 6 foods-12-02994-f006:**
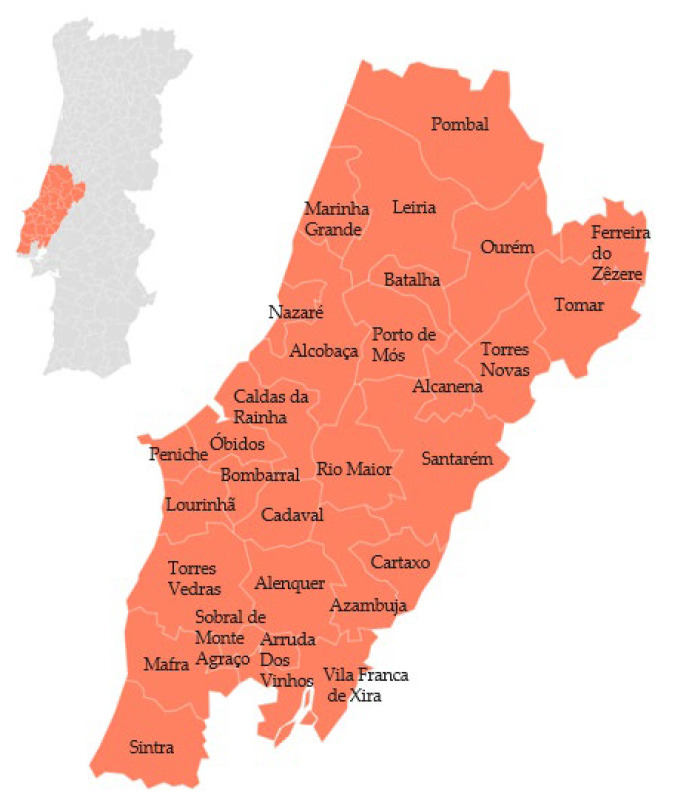
Geographical distribution of ‘Pera Rocha do Oeste’ PDO (pear). Adapted from [60].

**Figure 7 foods-12-02994-f007:**
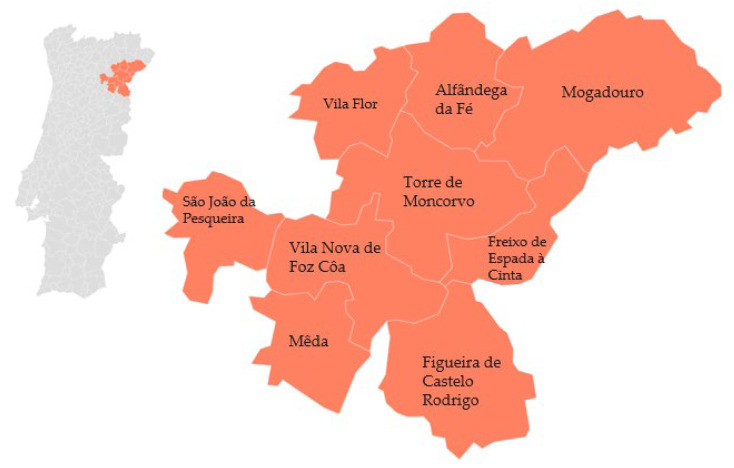
Geographical distribution of ‘Amêndoa Douro’ PDO (almond). Adapted from [68].

**Figure 8 foods-12-02994-f008:**
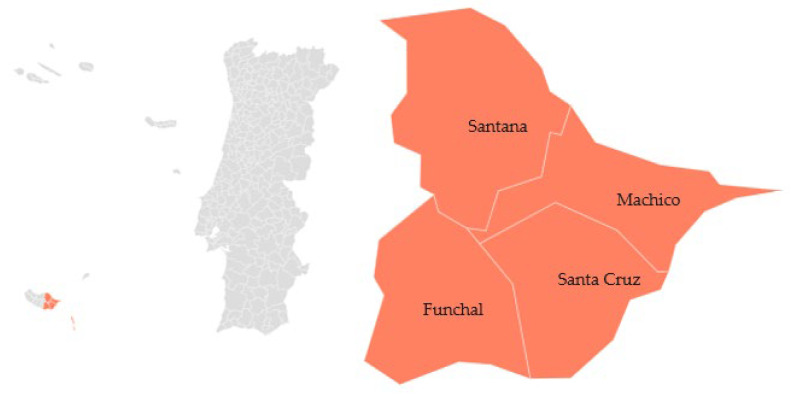
Geographical distribution of ‘Anona da Madeira’ PDO (cherimoya). Adapted from [73].

**Figure 9 foods-12-02994-f009:**
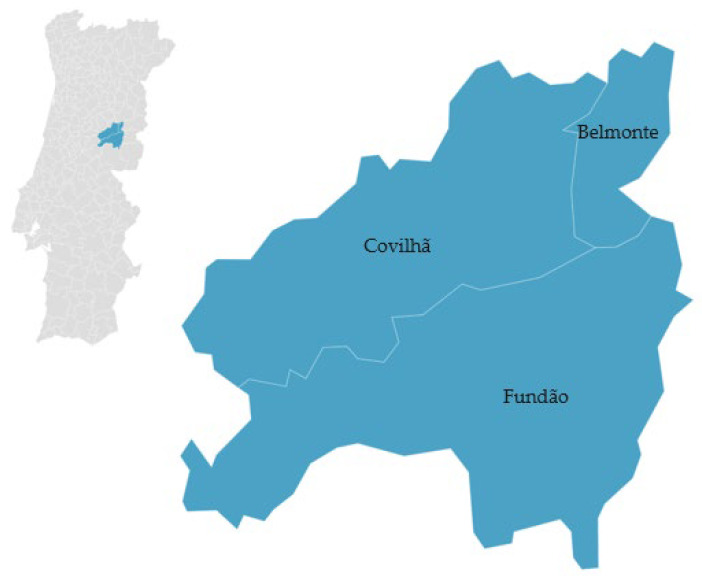
Geographical distribution of ‘Pêssego da Cova da Beira’ PGI (peach). Adapted from [80].

**Table 1 foods-12-02994-t001:** Portuguese products with PDO or PGI designation.

Portuguese Products	Denomination
‘Ameixa d’Elvas‘ (plum)	PDO
‘Amêndoa Douro‘ (almonds)	PDO
‘Ananás dos Açores/São Miguel‘ (pineaple)	PDO
‘Anona da Madeira‘ (cherimoya)	PDO
‘Azeitonas de Conserva de Elvas e Campo Maior‘(canned olives)	PDO
‘Azeitona de Conserva Negrinha de Freixo‘(canned olives)	PDO
‘Castanha da Padrela‘(chestnuts)	PDO
‘Castanha da Terra Fria‘(chestnuts)	PDO
‘Castanha dos Soutos da Lapa‘(chestnuts)	PDO
‘Castanha Marvão-Portalegre‘(chestnuts)	PDO
‘Cereja da Cova da Beira’ (cherries)	PGI
‘Cereja de São Julião-Portalegre‘ (cherries)	PDO
‘Cereja do Fundão‘ (cherries)	PGI
‘Citrinos do Algarve‘ (citrus)	PGI
‘Ginja de Óbidos e Alcobaça‘ (*Prunus cerasus*, cv. ‘Galega‘)	PGI
‘Maçã Bravo de Esmolfe‘ (apple)	PDO
‘Maçã da Beira Alta‘(apple)	PGI
‘Maçã da Cova da Beira’ (apple)	PGI
‘Maçã de Alcobaça‘(apple)	PGI
‘Maçã de Portalegre‘(apple)	PGI
‘Maçã Riscadinha de Palmela‘ (apple)	PDO
‘Maracujá dos Açores/São Miguel‘ (passion fruit)	PDO
‘Meloa de Santa Maria—Açores‘ (melon)	PGI
‘Pera Rocha do Oeste‘ (pear)	PDO
‘Pêssego da Cova da Beira‘ (peach)	PGI

**Table 2 foods-12-02994-t002:** Total production and production of certified fruits in 2020.

Fruit	Area of Production (ha)	Total Production (Tons)	Exported Production (Tons)	Exported Value (k€)	Certified Production (%)	Certified Products
Cherry	6387	9241	1057	1546	2.99	‘Cereja Cova da Beira’;
						‘Cereja de São Julião-Portalegre’;
						‘Ginja de Óbidos e Alcobaça’ (sour cherry)
Table Olive	6090	20,171	18,574	ND	0.01	‘Azeitonas de Conserva de Elvas e Campo Maior‘;
						‘Azeitona de Conserva Negrinha de Freixo‘
Orange	17,221	355,284	161,259	128,463	0.01	‘Citrinos do Algarve’
Apple	14,313	286,075	68,200	40,833	17	‘Maçã Bravo de Esmolfe’;
						‘Maçã da Beira Alta’; ‘Maçã da
						Cova da Beira’; ‘Maçã de
						Alcobaça’; ‘Maçã de
						Portalegre’;
						‘Maçã Riscadinha de Palmela’
Pear	11,325	131,004	94,913	81,666	75.5	‘Pera Rocha do Oeste’

**Table 3 foods-12-02994-t003:** Antioxidant activity of apple varieties [24].

Apples	ORAC ^a^	HORAC ^b^
‘Bravo de Esmolfe’	1503 ± 48	796 ± 94
‘Malápio Fino’	2236 ± 56	1183 ± 157
‘Malápio da Serra’	1389 ± 116	679 ± 93
‘Pêro Pipo’	1277 ± 79	715 ± 104
‘Golden’	821 ± 39	360 ± 52
‘Starking’	1486 ± 75	816 ± 123
‘Fuji’	1065 ± 75	551 ± 71
‘Gala Galaxy’	761 ± 22	446 ± 45
‘Reineta Parda’	1533 ± 141	886 ± 105

^a^—Oxygen radical absorbing capacity assay. Results are expressed as µmol of Trolox equivalents/100 g. ^b^—Hydroxyl radical adverting capacity assay. Results are expressed as µmol of caffeic acid equivalents/100 g. This table is reproduced under license from Elsevier.

**Table 4 foods-12-02994-t004:** Total phenolic content and DPPH assays.

Apple Variety	Total Phenolic (GAE µg/g)	DPPH (GAE µg/g)
‘Bravo de Esmolfe’ (PDO)	114	13
‘Golden Delicious’ (Beira Alta PGI)	42	3
‘Starking’ (Beira Alta PGI)	59	6

**Table 5 foods-12-02994-t005:** Characterization of ‘Bravo de Esmolfe’ apple extracts.

	In the Extract	In the Apple ^a^
HOSC (μmol CAET/g)	26.9	6.43
HORAC (μmol CAEAC/g)	18.4	4.40
ORAC (μmol CAET/g)	116.9	27.94
CAA (μmol de EQ/mg)	0.581	0.14
TP (mg GAE/g)	3.8	0.91
Flavonoids (mg/g)	1.17	0.28
(+)-Catechin (mg/g)	0.024	0.01
Chlorogenic acid (mg/g)	0.056	0.01
Procyanidin (mg/g)	0.131	0.03
(−)-Epicatechin (mg/g)	0.144	0.03
Quercetin-3′-glucoside (mg/g)	0.113	0.03
Quercetin-4′-glucoside (mg/g)	0.103	0.02

HOSC (Hydroxyl Radical Scavenging Capacity)—in micromoles of Trolox equivalents per gram (μmol CAET/g); HORAC in micromoles of ascorbic acid equivalents per gram (CAEAC/g); ORAC in μmol CAET/g; CAA (Cellular Antioxidant Activity) in micromoles of quercetin equivalents per milligram (μmol EQ/mg); TP (Total Phenolics) (Gallic Acid Equivalents per gram). ^a^—The content of bioactive compounds was calculated considering the reported extract yield of 23.9% [30].

**Table 6 foods-12-02994-t006:** Phenolic characterization and antioxidant activities of cherry traditional varieties.

	ORAC (μmol CAET/g)	HORAC (μmol of Caffeic Acid Equivalents/g)	TPC (mg GAE/100 g)	TAC ^a^ (mg of Cyanidin-3-Glucoside Equivalents/100 g)
‘Early Van Compact‘	120	64	956	224
‘Garnet‘	66	49	653	69
‘Lapin‘	177	93	1309	372
‘Maring‘	104	65	833	151
‘Morangão‘	66	44	555	5.6
‘Saco’	172	132	1309	296
‘Summit‘	50	24	440	28.8
‘Ulster‘	156	103	1187	292
‘Van‘	122	67	999	251

^a^—TAC, total anthocyanin content (expressed as mg of cyanidin-3-glucoside equivalents/100 g). This table is reproduced under license from Elsevier.

**Table 7 foods-12-02994-t007:** Total phenol and antioxidant capacity of olive seeds from ‘Cobrançosa’, ‘Galega vulgar’, and ‘Picual’ from a certified olive grove located in Elvas.

	TPC (mg GAE/g)	ABTS (µmol TE/g)	DPPH (µmol TE/g)
‘Cobrançosa’	11.9	54.03	3.64
‘Galega vulgar’	14.71	64.73	15.93
‘Picual’	13.03	57.42	6.1

**Table 8 foods-12-02994-t008:** Antioxidant activity of citrus varieties.

Orange Varieties/Location	TEAC (TE/L) ^a^	ORAC (TE/L) ^b^
‘Lanelate’ Faro	1.45	1.40
‘Lanelate’ Silves	1.51	1.65
‘Navelate’ Faro	1.40	------
‘Navelate’ Silves	1.75	3.40
‘Ortanique’ Faro	0.85	1.40
‘Ortanique’ Silves	0.90	0.75
‘Valencia late’ Faro	2.00	1.45
‘Valencia late’ Silves	1.70	1.45
‘D.João‘ Tavira	1.51	1.25
‘Encore’ Faro	0.52	3.45
‘Encore’ Silves	0.30	2.8

^a^—Trolox equivalent antioxidant capacity. Results are expressed as mol of TE/L from the juice of the different varieties cultivated in the Algarve. ^b^—Oxygen radical absorbing capacity assay. Results are expressed as mol of TE/L.

**Table 9 foods-12-02994-t009:** DPPH Assay and detected phenols in ‘Rocha’, ‘Comice’, ‘Abate’, and ‘General Lecler’ pears varieties.

Pear Variety	‘Rocha’	‘Comice’	‘Abate’	‘General Leclerc’
DPPH Assay (IC50, mg/mL)	0.11	0.133	0.22	0.37
Arbutin ^a^	22.5	5.6	3.6	2.6
gallic acid ^a^	4.4	5.5	7.6	6.3
(+)-catechin ^a^	2.9	1.2	2.4	6.9
chlorogenic acid ^a^	62.4	4.3	7.9	4.3
caffeic acid ^a^	11.1	3.5	12.9	8.5
syringic acid ^a^	24.7	5.4	-	8.5
(−)-epicatechin ^a^	3.8	-	-	-
coumaric acid ^a^	9.2	-	4.8	3.6
ferulic acid ^a^	23.3	-	9.6	4.8

^a^—in mg/100 g.

**Table 10 foods-12-02994-t010:** Total phenolic compounds (TPC) and antioxidant capacity (AOC) for the pulp and peel of pears in the fresh state and after drying.

Location	Torres Vedras	Azueira	Bombarral	Cadaval	Caria
Fresh Pulp (TPC ^a^)	291.0	264.6	239.2	272.5	228.1
Dried Pulp at 60 °C (TPC ^a^)	298.5	306.6	343.2	274.1	230.6
Fresh Pulp (AOC ^b^)	2261.0	2391.8	1802.2	1960.7	1710.0
Dried Pulp at 60 °C (AOC ^b^)	1255.9	1326.6	1170.6	1361.5	1331.7
Fresh Peel (TPC ^a^)	984.3	921.4	1032.9	981.0	782.1
Dried Peel at 60 °C (TPC ^a^)	723.3	641.6	644.6	541.4	479.3
Fresh Peel (AOC ^b^)	6617.2	7309.1	8007.9	7702.5	4748.7
Dried Peel at 60 °C (AOC ^b^)	3101.4	3045.0	2627.9	3331.2	2227.0

^a^—mg GAE/100 g ^b^—µmol TEAC/100 g.

## Data Availability

The data used to support the findings of this study can be made available by the corresponding author upon request.

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
