# Peer review of "Bioactive Compounds of Portuguese Fruits with PDO and PGI"

_foods, 2023, doi:10.3390/foods12162994_

Round 1
Reviewer 1 Report
The article describes the study comprehensively explored the properties of Bioactive compounds of Portuguese fruits with Protected Designation of Origin and Protected Geographical Indication. This review summarizes studies focusing on the bioactive compounds present in these fruits, an important factor in healthy foods for which of producers and consumers are becoming increasingly aware, contributing to development a high value bioeconomy. I think the subject is overall interesting. However, the manuscript needs comprehensive changes for improvement, especially in grammar. The review should be comprehensive and not be too long. The article needs major revisions as given below.
Abstract: Abstract should be comprehensive including all components of the review
Line 9-13: Rewrite the sentences for clarity
Line 14-15: Rewrite the sentences for clarity
Line 29: Add space before citations
Line 108: Add space before citations
Line 113, 196: Remove colored full stop
LINE 151: Add space before citations
Line 239: Remove double citations
Line 244: Remove double citations
Line 246-250: Remove double citations
Line 264: Remove double citations
Line 270: Remove double citations
Line 324: Add space before citations
Line 332: Add space before citations
Line 348: Add space before citations
Line 360: Remove double citations
Line 368: Remove double citations
Line 409-413: Rewrite the sentence for clarity
Line 422: Remove double citations
Line 429: Add space before citations
Line 453: Remove double citations
Line 458: Remove double citations
Line 463-465: Rewrite the sentence for clarity
Line 480, 482: Remove double citations
Line 513: Add space before citations
Line 518: Add space before citations
Line 530, 535: Add space before citations
Line 593: Table should be place after line 593
Line 623: Add space before citations
Line 652: Remove double citation
Line 681, 684: Remove double citation
Line 718: Remove double citation
Line 727, 728: Remove double citation
Line 734: Remove double citation
Line 766, 770: Add space before citations
References should be in same style following journal guide line (847, 848, 863: The style of references is different from others)
Provide details of these references: Line 1028, 1029
Author Response
Please see the attachment.
Thank you,
Catarina Nunes.

Reviewer 2 Report
In the present manuscript, the situation of the bioactive compounds of fruits produced in Portugal with PDO and PGI was reviewed. The authors did a nice work to give us a useful summary. However, though we knew this work was really hard, there were quite a lot of type errors in this manuscript. Furthermore, some expression in this manuscript was not correct, which gave rise to misunderstanding sometimes. Besides, we thought the authors omitted the most important fruits, grapes (besides olives), we hope they can add this part to the revised manuscript, or give us an acceptable explanation, why? All in all, we gave a major revision to this manuscript. Some comments or suggestions were as following:
1, Line 119-Line 123, there should be no blank space between the number and %.
2, Line 150-Line 152, the introduction of the bioactive compounds was not clear. For example, polyphenols, tannins, flavonoids and flavanols were not at the same level. Flavanols was one class chemicals of flavonoids, which was also a class of compounds of polyphenols. All tannins were polyphenols, but condensed tannins, also named as proanthocyanidins, belonged to flavanols. Also, anthocyanin was one class of plant pigments. It was also a kind of flavonoids. Thus, such expression should be revised.
3, Line 154, the sentence “polyphenols are responsible for the color of the fruits” was not correct, since not all the polyphenols were colored, except for anthocyanins. Besides, there were also other kinds of plant pigments in fruits, for example, chlorophylls, carotenoids, betacyanins, which were also responsible for the color of different fruits.
4, Line 174-Line 175, the introduction of flavonoids was not correct. All the flavonoids had the structures of three rings of C6-C3-C6, but not only the two aromatic rings. Besides, not all the flavonoids had glucosides.
5, Line 208, Line 397, unify the terms of (+)-catechin and (-)-epicatichin.
6, Line 230, the letter “O” in “phloretin-2-O-xyloglucoside” should be in italic. There were also quite a lot of similar type errors in the following text.
7, Line 308, when the genus name “Prunus” occurred more than once in the same paper, it should be abbreviated as “P.”. There were the same problems about Citrus.
8, Line 350, the letter “p” in “p-coumaric” should be in italic.
9, Line 366, the variety name Saco should be enclosed in single quotation marks as ‘Saco’, but not the double ones.
10, Line 368, the reference [16] occurred twice at the end of the sentence.
11, Line 356, the initial English letter “C” of “5-Caffeoylquinic acid” should be lowercase. Similar errors were in “p-Hydroxybenzoic acid (Line 588)” and “p-Coumaric acid”.
12, Line 531, “50-100%” should be revised as “50%-100%”.
13, There were no part about grapes in this manuscript, which had the largest production in Portugal (besides olives) as fruits and quite important for their economy. We hope the authors could add this part to the paper or explain why.
14, The formats of some references were not correct. For example, [35] was incomplete.
No
Author Response

(The authors gave the same response as above.)

Reviewer 3 Report
Dear Authors,
Detailed notes on the manuscript are given below:
1) in the final part of the chapter "Introduction" clearly state the purpose of the work (the purpose of the work was ...),
2) Figures 1-9; in my opinion gray background + black font and its size = not suitable (lighten the background, increase the font size),
3) Chapter 3. Conclusions and future perspectives; this chapter is a summary (not traditional conclusions), I suggest changing it to "Summary and future perspectives".
Author Response

(The authors gave the same response as above.)

Round 2
Reviewer 2 Report
Accept, no more comments.